# Incoherent Region-Aware Occlusion Instance Synthesis for Grape Amodal Detection

**DOI:** 10.3390/s25051546

**Published:** 2025-03-02

**Authors:** Yihan Wang, Shide Xiao, Xiangyin Meng

**Affiliations:** School of Mechanical Engineering, Southwest Jiaotong University, Chengdu 610031, China; wyh123@my.swjtu.edu.cn (Y.W.); sdxiao@home.swjtu.edu.cn (S.X.)

**Keywords:** grape segmentation, amodal detection, self-supervision learning, instance synthesis

## Abstract

Occlusion presents a significant challenge in grape phenotyping detection, where predicting occluded content (amodal detection) can greatly enhance detection accuracy. Recognizing that amodal detection performance is heavily influenced by the segmentation quality between occluder and occluded grape instances, we propose a grape instance segmentation model designed to precisely predict error-prone regions caused by mask size transformations during segmentation, with a particular focus on overlapping regions. To address the limitations of current occlusion synthesis methods in amodal detection, a novel overlapping cover strategy is introduced to replace the existing random cover strategy. This approach ensures that synthetic grape instances better align with real-world occlusion scenarios. Quantitative comparison experiments conducted on the grape amodal detection dataset demonstrate that the proposed grape instance segmentation model achieves superior amodal detection performance, with an IoU score of 0.7931. Additionally, the proposed overlapping cover strategy significantly outperforms the random cover strategy in amodal detection performance.

## 1. Introduction

Phenotypic detection is integral to vineyard management, facilitating real-time monitoring of grape development and yield estimation, which are essential for guiding agronomic practices and optimizing commercial decision making [1,2,3,4,5]. Leveraging the benefits of non-destructive and high-density sampling, visual detection methods have increasingly supplanted traditional manual sampling, establishing themselves as the primary approach for large-scale phenotypic evaluation in vineyards while achieving significantly enhanced detection accuracy [5,6,7,8,9]. Nonetheless, occlusion remains a persistent challenge, always leading to notable underestimation in visual-based grape phenotyping outcomes and serving as the primary source of detection deviations [1,5].

The process of predicting the complete content or contour of partially occluded objects, commonly referred to as amodal detection or completion, has wide-ranging applications in areas such as grasp localization [10], autonomous driving [11,12], and harvest position perception [13]. While shape priors are often employed as a foundational approach for amodal contour completion [12,14], they are ill suited for objects like grapes due to their highly variable and irregular contours. Amodal detection face unique challenges compared to other visual detection tasks, primarily due to the difficulty of obtaining paired images that include both occluded objects and their corresponding intact content under real-world conditions, making it impossible to annotate occluded objects. Additionally, the irregular and diverse shapes of grapes render the manual generation of amodal annotations particularly unreliable. In the absence of amodal annotations, building deep learning-based image mapping models becomes an exceedingly complex task.

Recent studies have primarily addressed amodal detection and completion in the absence of real occlusion annotations by leveraging synthesized occlusions [1,15,16]. These methods typically synthesize occlusion scenarios by combining readily collectible complete instances, thereby establishing a mapping between occluded and complete instances. Therefore, the fidelity of synthesis occlusion instances is very important to the performance of amodal detection. Furthermore, the potential complete region of an occluded instance cannot be accurately predicted beforehand; the only prior information is that the occluded region resides within the spatial boundary of the occluding instance. Consequently, amodal detection paradigms often require pairing occluding and occluded instances during the prediction process [15,16], making amodal detection performance highly sensitive to instance segmentation results, particularly in overlapping boundary regions. As shown in Figure 1a, the slight segmentation gaps between the occluding and occluded instances can significantly impact the performance of amodal detection. Figure 1b reports significant differences in amodal detection performance between segmentation results and annotations. Further visual comparisons indicate that this performance gap is primarily attributed to the segmentation quality of boundaries at overlapping regions between occlude and occluder instances.

Instance segmentation is a key technique in grape phenotyping detection, extensively used in recent years for extracting targets such as grapevines [7], clusters [9,17], and berries [18]. To improve segmentation performance in unstructured environments, various improvements in segmentation methods, including attention mechanisms [9] and backbone improvements [8,18], have been proposed. However, these improvements often neglected the overlapping regions between instances. The high semantic similarity within overlapping areas of grape clusters often leads to blurry and misaligned predictions. Recent studies have aimed to improve instance segmentation qualities by refining coarse results through additional optimization networks. For example, PointRend [19] enhances segmentation by iteratively predicting labels for pixels at object edges or regions with uncertain labels within the region of interest. Similarly, Tang et al. [20] improved segmentation accuracy by post-processing instance boundary image patches along with their corresponding coarse masks. Ke et al. [21] observed that the small RoI pooling size in Mask R-CNN and its variants inherently results in coarse mask predictions due to spatial information loss caused by downsampling operations. To address this, they proposed a cascaded transformer network specifically designed to handle error-prone regions during the sampling process. Additionally, recognizing that errors in coarse predictions frequently occur at instance boundaries, Cheng et al. [22] introduced an explicit boundary prediction branch to provide supplementary guidance, significantly enhancing mask prediction accuracy. However, these methods appear to focus solely on instance boundaries and overlook the overlapping regions between instances. Although Ke et al. [23] proposed a Bilayer Convolutional Network to separately predict occluder and occluded instances, thereby improving the boundary segmentation quality in overlapping regions, this method requires distinguishing between occluder and occluded instances within the dataset.

As shown in Figure 2, our objective is to achieve amodal detection for grapes without relying on amodal annotations. To address this challenge, we propose a grape instance segmentation model designed to accurately delineate grape instances from proximal images. These instances are then paired based on the presence of occlusion, and a grape amodal detection model is introduced to achieve amodal detection for the paired instances. In the grape instance segmentation model, error-prone regions (incoherent region) caused by size transformations during segmentation training are identified and utilized to guide the model toward achieving more accurate segmentation of overlapping grape area. For the grape amodal detection model, a novel overlapping cover strategy is proposed to replace the traditional random cover approach [1,15,16], ensuring the realism of synthetic occluded instances. Compared with other segmentation methods, our grape instance segmentation model achieves superior amodal detection performance with grape amodal detection datasets. The overlapping cover strategy can further improve model’s amodal detection capability.

To advance research in grape amodal detection, we have open-sourced the related datasets and code. The source dataset is available at https://doi.org/10.5281/zenodo.14630616, accessed on 2 July 2023. The source code is available at https://github.com/YihanWang-chengdu/grape_de_occlusion (github.com, accessed on 2 July 2023). Overall, our main contributions are as follows:(1)We propose a self-supervision grape amodal detection pipeline that enables amodal detection for overlapping grapes.(2)The proposed grape instance segmentation model offers improved accuracy in segmenting grape overlapping regions, providing a stronger foundation for subsequent amodal detection.(3)Qualitative and quantitative comparisons consistently demonstrate our overlapping cover synthesis strategy’s effectiveness in enhancing grape amodal detection performance.

## 2. Methods

### 2.1. Process Description

The goal of our work is to predict the potential complete contours of occluded grape instances (amodal detection) in grape proximal images, even in the absence of amodal annotations. As shown in Figure 2a, for a given proximal grape image I, a grape instance segmentation model is first developed to generate segmentation masks for all grape instances $M^{n}$. Next, these instances are paired based on whether their bounding boxes overlap $\left\{M^{i},M^{j}|i,j\in n\right\}$. A trained grape amodal detection model is then applied to cross-predict the amodal masks for these instance pairs: $f(M^{i},M^{j})\Rightarrow M_{a}^{i}$, $f(M^{j},M^{i})\Rightarrow M_{a}^{j}$. We use $\Delta_{M}$ to represent the proportion of mask area increase before and after amodal detection. The occlusion relationship within this instance pair is determined based on the following rule.(1)$$\begin{array}{l} \Delta_{M}=(M_{a}-M)/M\\ O(M^{i},M^{j})=\left\{\begin{array}{l} \begin{array}{cc} -1 & \mathrm{if}\;\Delta_{M^{i}},\Delta_{M^{j}}=0 \end{array}\\ \begin{array}{cc} 1 & \mathrm{if}\;\Delta_{M^{i}}>\Delta_{M^{j}} \end{array}\\ \begin{array}{cc} 0 & \mathrm{if}\;\Delta_{M^{j}}>\Delta_{M^{i}} \end{array} \end{array}\right. \end{array}$$

Here, $O(M^{i},M^{j})=-1$ indicates that there is no overlap between two instances. $O(M^{i},M^{j})=1$ indicates that $M^{j}$ occluded $M^{i}$, and $O(M^{i},M^{j})=0$ represents the inverse occlusion relationship.

### 2.2. Training of the Grape Instance Segmentation Model

Mask-RCNN [24] family instance segmentation methods require sampling of annotation masks for training supervision, which inherently leads to the loss of spatial information (referred to as incoherent regions; Figure 3). This issue is especially pronounced in overlapping areas, where critical structural details are often compromised. Therefore, we propose incorporating an additional decoder into the segmentation model, equipped with larger ROI pooling and output sizes, to focus specifically on these regions by preserving sufficient feature details and minimizing spatial information loss during size transformations.

The primary goal is to extract instance masks for each proposal, as well as instance-level incoherent region masks and local incoherent region masks (the incoherent region in the overlapping instance region) generated under various scale transformations during model training. Therefore, a paradigm for capturing incoherent region masks resulting from a single size transformation is introduced, as defined in (2).(2)$$M_{\mathrm{Inc}}=D(m⊕U(D(m)))$$

Here, m is input mask, U and D represent the bilinear interpolation operations for upsampling and downsampling, respectively, while $M_{\mathrm{Inc}}$ represents the incoherent region mask generated at the current size transformation.

To balance computational efficiency and accuracy, the grape proximal images are typically resized to align with input dimensions (often smaller) of the grape instance segmentation model. Utilizing (2), the incoherent mask $M_{\mathrm{Inc}}$ resulting from this size transformation is derived from the instance annotations $M_{\mathrm{In}}$. Non-zero points $i_{\left(x,y\right)}$ in the incoherent mask $M_{\mathrm{Inc}}$ are transversed, and for each point, three pixel pairs are sampled. If at least two pixel pairs are located at different instances, this point will be classified as part of local incoherent region mask $M_{\mathrm{Inc}-o}$.(3)$$\begin{array}{l} n=\sum^{3}1_{\left[p_{1}\neq p_{2}\right]}\\ \left\{\begin{array}{l} i_{\left(x,y\right)}\in M_{\mathrm{Inc}-o},n\geq 2\\ i_{\left(x,y\right)}\notin M_{\mathrm{Inc}-o},\;otherwise \end{array}\right. \end{array}$$
where 1 represents the Indicator function. As shown in Figure 4a, based on the proposals generated by instance segmentation network, $M_{\mathrm{Inc}-o}^{C}$ and $M_{P}^{C}$ are cropped from the local incoherent region mask $M_{\mathrm{Inc}-o}$ and the corresponding instance mask $M_{p}$. To drive the training of network, the size of the supervision masks must align with the model’s requirements. Therefore, applying (2) obtains the local incoherent mask $M_{\mathrm{Inc}-o}^{\prime }$ and instance incoherent mask $M_{p}^{I}$ at this size transformation from $M_{\mathrm{Inc}-o}^{C}$ and $M_{P}^{C}$. Finally, applying a logical AND operation between $M_{\mathrm{Inc}-o}^{\prime }$ and $M_{p}^{I}$ obtains the instance local incoherent region mask $M_{\mathrm{Inc}-o}^{I}$.

Figure 5a presents the structure of the grape instance segmentation model. Similar with Mask-RCN [24], our grape instance segmentation model comprises an encoder, a region proposal network (RPN), and multiple detector heads. The encoders, consisting of a backbone network and a Feature Pyramid Network, is responsible for extracting feature representations at various scales. The region proposal network generates region proposals for areas of interest, while box and mask decoders are used to classify and segment instances within these proposals. Furthermore, an incoherent region decoder is imbedded to predict incoherent regions. Figure 5b presents the detailed architectures of the customized mask decoder and the incoherent region decoder.

For RoI Align features of the mask decoder and incoherent region decoder, $F_{M}$ and $F_{I}$, with resolutions of 14 × 14 and 28 × 28 pixels, multiple cascaded convolutional layers are applied to refine them. Subsequently, in the mask decoder, a deconvolution operation is performed to upsample the feature maps, aligning their resolution with that of the incoherent region decoder’s features. For aligned features, a pixel affinity learning module is employed to model the feature correlation. Following this, a 3 × 3 convolutional layer with a stride of 2 is applied to downsample both decoder features. An additional pixel affinity learning module is then applied to further model the pixel correlation of two decoder features at different scales. Finally, deconvolution layers are applied to upsample the features from both decoders, generating the final predictions $P_{m}$ and $P_{I}$ with resolutions of 28 × 28 and 56 × 56, respectively.

For the decoder predictions $P_{m}$ and $P_{I}$, the instance mask $M_{p}^{C}$, the instance incoherent region mask $M_{p}^{I}$, and the local incoherent region mask $M_{\mathrm{Inc}-o}^{I}$ are the supervisory signals driving the training of grape instance segmentation model. The loss function $L_{M}$ for the mask decoder is defined as follows:(4)$$L_{M}=L_{\mathrm{BCE}}(P_{m},M_{p}^{C})$$
where $L_{\mathrm{BCE}}$ represents the binary cross-entropy (BCE) loss. The loss function $L_{I}$ for the incoherent region decoder is defined as follows:(5)$$L_{I}=L_{\mathrm{DCIE}-\mathrm{BCE}}(P_{I},M_{p}^{I})+\lambda L_{\mathrm{DCIE}-\mathrm{BCE}}(P_{I}\cap M_{\mathrm{Inc}-o}^{I},M_{\mathrm{Inc}-o}^{I})$$
where $P_{I}\cap M_{\mathrm{Inc}-o}^{I}$ represents the intersection between the incoherent region decoder predictions $P_{I}$ and the instance local incoherent region mask $M_{p}^{I}$. The λ represents the weight parameter, which is set to 2 during the training process. The $L_{\mathrm{DCIE}-\mathrm{BCE}}$ is defined as the weighted sum of the dice loss [22] and the BCE loss.(6)$$\begin{array}{l} L_{\mathrm{DCIE}-\mathrm{BCE}}=L_{\mathrm{DICE}}+L_{\mathrm{BCE}}\\ L_{\mathrm{DICE}}(P,M)=1-\frac{2\sum_{i}P^{i}M^{i}+1}{\sum_{i}{\left(P^{i}\right)}^{2}+\sum_{i}{\left(M^{i}\right)}^{2}+1} \end{array}$$

Given that incoherent region masks are a subset of instance masks, there is a significant pixel-level affinity between them [25]. Therefore, we integrate the pixel affinity learning module, which enables the two decoders to collaborate effectively and ensures consistent predictions between them. The structure of the pixel affinity learning module is depicted in Figure 6b.

Let $F_{M}^{\prime }$ and $F_{I}^{\prime }$ represent the input features of the pixel affinity learning module. These features are first concatenated along the channel dimension to form a unified feature representation. The concatenated feature is then processed by a nonlinear mapping unit, which consists of stacked convolutional blocks and an activation function. The unit compresses the features from $R_{h\times w\times 2c}$ space to $R_{h\times w\times c}$ space and subsequently maps them back to $R_{h\times w\times 2c}$, achieving feature compression and excitation. The above process can be represented as follows:(7)$$F=\Phi ([\mathrm{concat}[F_{M}^{\prime },F_{I}^{\prime }])$$
where concat and Φ(•) represent the concatenation operation and nonlinear mapping unit. Next, the features $F\in R_{h\times w\times 2c}$ are split along the channel dimension, and a SoftMax activation function is applied to compute the weight coefficients. These weight coefficients are subsequently used to reweight the input features.(8)$$F_{m}=[F_{M}^{\prime },F_{I}^{\prime }]\;⊗\mathrm{SoftMax}([F_{1},F_{2}])$$
where ⊗ represents the element-wise multiplication operation. Following the above, the reweighted features $F_{m}\in R_{h\times w\times c\times 2}$ are concatenated along the channel dimension and passed through a convolutional block to reduce its channel dimension, resulting in a shared representation of the two decoder features $F_{s}\in R_{h\times w\times c}$.(9)$$\begin{array}{l} F_{M}^{\prime \prime }=SE(F_{S})+F_{M}^{\prime }\\ F_{I}^{\prime \prime }=SE(F_{S})+F_{I}^{\prime } \end{array}$$

For the shared representation $F_{s}$, Equation (9) reports the output features of the pixel affinity learning module. Here, SE denotes the attention module [26], and + represents the element-wise addition operation.

Although the visual results indicate that incoherent regions are primarily located near boundary areas, there is a significant distinction between the incoherent regions and boundary supervision used in [22]. As present in Figure 4b, boundary masks are generally narrower and more prone to irregularities, will negatively impact segmentation performance when used as supervision. This issue becomes especially evident when the supervision size is small, resulting in unsmooth and distorted boundary masks that lack the robustness observed in incoherent masks.

### 2.3. Training of the Grape Amodal Detection Model

Beyond being affected by segmentation quality, the performance of amodal detection also significantly depends on how closely the synthesized occlusion domain corresponds to the real-world occlusion domain. Current approaches often rely on random cover strategies [1,15,16] to synthesize occluded instances, failing to capture the authentic occlusion patterns observed in grapes. Therefore, we propose a novel overlapping cover strategy designed to replicate the natural occlusion patterns of grapes.

Suppose that the pose of grape occlusion takes a value ranging from 0° to 360°, and 45° per interval refers to an occlusion pattern, as shown in Figure 6. Accordingly, we define the occlusion pattern with a total of 8 cases. Taking Case #2 as an example, we demonstrate the synthesis process using the overlapping cover strategy, as outlined in Algorithm 1. The outline of the Case #2 instance synthesis process is as follows:An instance pool with 2427 complete grape instances $\left\{I^{r}\right\}$ manually annotated and cropped from proximal images serves as a source of diverse instances.Instances from the instance pool are randomly selected as occluded instance $I_{\mathrm{Occ}}^{r}$ and occluder instance $I_{\mathrm{Occder}}^{r}$.A randomly initialized occlusion parameter β is employed to control the dimensions $w_{\beta }$ and $h_{\beta }$ of the overlapping area between two instances.Based on $w_{\beta }$ and $h_{\beta }$, the overlapping operation is applied and the content of occluded instance in the overlapping area is erased. The minimal bounding rectangle image containing the two instances is constructed, filled accordingly, and outputs the synthesized result $I_{Occ}^{s}$, $I_{\mathrm{Intact}}^{s}$, $I_{\mathrm{Occder}}^{s}$.


**Algorithm 1:** The grape occlusion synthesis process in Case #2

$\mathrm{Input}:\;\mathrm{The}\;\mathrm{Occluded}\;\mathrm{instance}\;I_{\mathrm{Occ}}^{r}$


$,\;\mathrm{which}\;\mathrm{is}\;\mathrm{randomly}\;\mathrm{selected}\;\mathrm{from}\;\mathrm{complete}\;\mathrm{instance}\;\mathrm{set}\;\left\{I^{r}\right\}$



$\mathrm{The}\;\mathrm{width}\;\mathrm{and}\;\mathrm{height}\;\mathrm{of}\;\mathrm{the}\;\mathrm{instance}\;w_{occ}$


$,\;h_{\mathrm{occ}}$



Occlusion pattern α=2



$\mathrm{Initialize}\;\mathrm{the}\;\mathrm{occlusion}\;\mathrm{ratio}\;\mathrm{parameters}\;\beta \in \left\{x\in R|x\in (\begin{array}{cc} 0.2 & 0.85 \end{array})\right\}$



$\mathrm{Initialize}\;\mathrm{the}\;\mathrm{reverse}\;\mathrm{synthesis}\;\gamma \in \left\{\mathrm{True},\mathrm{False}\right\}$



$\mathrm{Output}:\;\mathrm{The}\;\mathrm{occluded}\;\mathrm{synthesis}\;\mathrm{image}\;I_{Occ}^{s}$



$\mathrm{The}\;\mathrm{occluder}\;\mathrm{synthesis}\;\mathrm{image}\;I_{\mathrm{Occder}}^{s}$



$\mathrm{The}\;\mathrm{intact}\;\mathrm{image}\;\mathrm{corresponds}\;\mathrm{occluded}\;\mathrm{image}\;I_{\mathrm{Intact}}^{s}$

1. $\mathrm{While}\;\mathrm{select}\;I_{\mathrm{Occder}}^{r}$$\;\mathrm{in}\;\left\{I^{r}\right\}$:2. $\mathrm{Obtain}\;\mathrm{the}\;\mathrm{dimension}\;\mathrm{parameters}\;\mathrm{of}\;I_{\mathrm{Occder}}^{r}$$\;\mathrm{instance}\;w_{occder}$ $h_{occder}$3. $\mathrm{If}\;h_{occ}\geq 1.2h_{occder}$$\;\mathrm{or}\;h_{occ}\leq 0.9h_{occder}$:4. $\;\;\mathrm{reselect}\;\mathrm{another}\;I_{\mathrm{Occder}}^{r}$$\;\mathrm{in}\;\left\{I^{r}\right\}$5. **else:**6.  SwitchCase =α=2:7. $\;\;\mathrm{Calculate}\;\mathrm{the}\;\mathrm{overlapping}\;\mathrm{area}\;\mathrm{parameters}\;w_{\beta }\leftarrow w_{occ}*\beta$$,\;h_{\beta }\leftarrow h_{\mathrm{occ}}*\beta$8. $\;\;\mathrm{Create}\;\mathrm{two}\;\mathrm{blank}\;\mathrm{images}I_{1}^{s}$$,\;I_{2}^{s}$ of the same dimension, matching the bounding rectangles

$\mathrm{of}\;\mathrm{the}\;\mathrm{occluding}\;\mathrm{and}\;\mathrm{occluded}\;\mathrm{instances},\;\mathrm{with}\;\mathrm{dimensions}\;\mathrm{specified}\;\mathrm{as}\;h_{s}=h_{occ}+h_{occder}-h_{\beta },$



$w_{s}=w_{occ}+w_{occder}-w_{\beta }$

9. $\;\;\mathrm{Fill}\;\mathrm{occluded}\;\mathrm{and}\;\mathrm{occluder}\;\mathrm{instances}\;\mathrm{into}\;\mathrm{the}\;\mathrm{two}\;\mathrm{blank}\;\mathrm{images}\;I_{1}^{s}[0:w_{occ}\;,\;h_{occder}-h_{\beta }:h_{s}]\leftarrow I_{\mathrm{Occ}}^{r}$

$,\;I_{2}^{s}[w_{occ}-w_{\beta }\;:\;w_{s}\;,\;0\;:\;h_{occder}]\leftarrow I_{\mathrm{Occder}}^{r}$

10.   Apply grayscale conversion to the image, followed by binarization, to generate its

$\mathrm{corresponding}\;\mathrm{mask},\;M_{\mathrm{occ}}=\mathrm{Bool}(\mathrm{Gray}(I_{2}^{s}))$

11.   If not γ:12. $\;\;\;\;I_{Occ}^{s}\leftarrow I_{1}^{s}*({\sim M}_{\mathrm{occ}})$$,\;I_{\mathrm{Intact}}^{s}\leftarrow I_{1}^{s}$$,\;I_{\mathrm{Occder}}^{s}\leftarrow I_{2}^{s}$13. $\;\;\;\;\mathrm{Calculate}\;\mathrm{occlusion}\;\mathrm{rate}\;\varepsilon =1-\frac{{\mathrm{count}(\mathrm{Bool}(\mathrm{Gray}(I}_{\mathrm{Occ}}^{s})))}{{\mathrm{count}(\mathrm{Bool}(\mathrm{Gray}(I}_{\mathrm{Intact}}^{s})))}$14.     While ε≤0.1:15.       β=β+0.05, Re-execute this **Case**16.  **else**:17. $\;\;\;\;\;I_{Occ}^{s}\leftarrow I_{2}^{s}$$,\;I_{\mathrm{Intact}}^{s}\leftarrow I_{2}^{s}$$,\;I_{\mathrm{Occder}}^{s}\leftarrow I_{1}^{s}*({\sim M}_{\mathrm{occ}})$18. $\;\;\;\;\;\;\mathrm{Calculate}\;\mathrm{occlusion}\;\mathrm{rate}\;\varepsilon =1-\frac{{\mathrm{count}(\mathrm{Bool}(\mathrm{Gray}(I}_{\mathrm{Occder}}^{s})))}{{\mathrm{count}(\mathrm{Bool}(\mathrm{Gray}(I}_{1}^{s})))}$19.         While ε≤0.1:20.           β=β+0.05, Re-execute this **Case**21. $\;\mathrm{Output}\;I_{Occ}^{s}$$,\;I_{\mathrm{Occder}}^{s}$$,\;I_{\mathrm{Intact}}^{s}$


For the synthesized instances $I_{Occ}^{s}$, $I_{\mathrm{Intact}}^{s}$, and $I_{\mathrm{Occder}}^{s}$, their respective binary masks are denoted as $M_{Occ}^{s}$, $M_{\mathrm{Occder}}^{s}$, and $M_{\mathrm{Intact}}^{s}$. The training process of our grape amodal detection model is guided using the supervision method specified in (10).(10)$$L_{\mathrm{amodal}}(f_{\mathrm{Unet}}(M_{Occ}^{s},M_{\mathrm{Occder}}^{s}),M_{\mathrm{Intact}}^{s})$$
where $f_{\mathrm{Unet}}$ represents the amodal detection network, which is primarily based on U-Net [15]. $M_{Occ}^{s}$ and $M_{\mathrm{Occder}}^{s}$ are concatenated along the channel dimension to serve as model input. For output prediction $p_{a}$ of the model $f_{\mathrm{Unet}}$, $L_{\mathrm{amodal}}$ is defined as follows:(11)$$\begin{array}{l} L_{\mathrm{amodal}}=L_{\mathrm{Bce}}((p_{a}\cap M_{\mathrm{Occder}}^{s}),(M_{\mathrm{Intact}}^{s}\cap M_{\mathrm{Occder}}^{s}))\\ +\lambda_{2}L_{\mathrm{Bce}}((p_{a}\cap {\sim M}_{\mathrm{Occder}}^{s}),(M_{\mathrm{Intact}}^{s}\cap {\sim M}_{\mathrm{Occder}}^{s})) \end{array}$$

Here, $L_{\mathrm{Bce}}$ represents the BCE loss, and $\lambda_{2}$ denotes the weight coefficient, which is typically set to 5 during training.

## 3. Dataset and Experimental Setting

### 3.1. Datasets

The grape proximal images in the grape instance segmentation and amodal detection datasets were collected at the vineyard of Sichuan Pumancang Agricultural Technology Co., LTD (30°26′27″ N, 103°82′59″ E) in Chengdu, Sichuan, China. The cultivated area covers approximately 278.82 acres and is exclusively dedicated to the Shine Muscat grape variety. The grapevines are supported by trellises positioned approximately 170–190 cm above the ground. We mounted an Intel RealSense L515 camera (Intel, Santa Clara, CA, USA) on a bracket attached to an AGV (Figure 7), positioning it approximately 150 cm above the ground. The AGV was remotely controlled to move forward at a distance of about 70 cm from the grapevine.

Table 1 provides detailed information about the grape instance segmentation and amodal detection datasets. The grape instance segmentation dataset consists of 2145 images, each with a resolution of 1280 × 720 pixels. Using LabelMe software (version 5.5.0), we annotated all foreground grape clusters (those belonging to the grapevine on the camera-facing side), resulting in a total of 12,636 instances. The dataset was split into 60% (1236) for training, 20% (406) for validation, and 20% (503) for testing of the various instance segmentation methods.

To rigorously evaluate the performance of grape amodal detection, we curated and annotated a dedicated grape amodal detection dataset comprising paired images: occluded grape images and their corresponding de-occluded counterparts. Image acquisition involved positioning the vehicle at a fixed location to capture the proximal image. Subsequently, the occluder grapes were manually removed, allowing for the capture of de-occluded images (Figure 7). Annotations were conducted using LabelMe software (version 5.5.0). In the occluded images, the occluded instance (having the corresponding complete instance in the de-occluded image) and their corresponding occluder instances were meticulously labeled. The complete instances from the de-occluded images were then annotated and copied back to the occluded images for comprehensive annotation consistency. It is noteworthy that this dataset was exclusively employed for benchmarking the amodal detection performance of the proposed method, without contributing to model training.

It is important to note that, in addition to capturing grape proximal images using an RGBD camera, the corresponding depth maps were also acquired. This is because obtaining the precise distance of the grape clusters from the camera can enhance the accuracy of phenotype detection. However, since the primary focus of this study is on amodal detection on RGB images, the depth map information is not discussed in detail in this paper.

### 3.2. Implementation Details and Metrics

The experimental apparatus for this study is a Dell Precision 5820 Tower (Dell Technologies Inc., Round Rock, TX, USA) workstation running Ubuntu 20.04. The workstation features an Intel i9-10900X CPU (Intel Technologies Inc., Santa Clara, CA, USA), 64 GB of RAM, and a GeForce RTX 3090 GPU (NVIDIA Corporation, Santa Clara, CA, USA) with 24 GB of RAM. The training and testing of all experiment codes are conducted under the environment with PyTorch 1.10.1 and CUDA 11.3.

The proposed grape instance segmentation model, along with several instance segmentation algorithms used for comparison, was implemented using the popular instance segmentation library Detectron2 (version 0.6) [27]. Notably, SOLOv2 [28] and YOLACT++ [29] were executed within the MMDetection framework [30]. The backbone networks of all the segmentation algorithms were ResNet50 and were initialized with ImageNet pretrained weights, and all batch normalization (BN) layers within them were unfrozen. To make a trade-off between computational efficiency and segmentation accuracy, the grape images were initially scaled down to 875 × 432 × 3 pixels (60% of the original size) before being fed into segmentation network. We set the initial learning rate at 0.0001 and adopted Adam as the optimizer for training. Warm-up and cosine annealing were employed to adjust the learning rate. All networks used a single GPU for universal training in a batch size of 4.

Following the evaluation protocol for instance segmentation, we trained the networks under the training set of grape instance segmentation dataset for 60 epochs. The COCO average precision (COCO AP) and average precision for boxes (**AP^Box^**) were reported on the test set, where COCO AP is the average precision over different IoU thresholds (from 0.5 to 0.95). **AP^Box^** was computed by calculating the average precision at various intersection-over-union (IoU, from 0.5 to 0.95) thresholds for bounding boxes.

Considering the impact of boundary segmentation performance on amodal detection, we followed the study of [31], which adopts boundary average precision AP^B^ to evaluate the boundary quality of segmentation results, which is the average precision over different boundary IoU thresholds (from 0.5 to 0.95). For prediction mask P and ground truth mask G, the boundary IoU is defined as follows:(12)$$\mathrm{BoundaryIoU}\;(G,\;P)=\frac{\left|(G_{d}\cap G)\cap (P_{d}\cap P)\right|}{\left|(G_{d}\cap G)\cup (P_{d}\cap P)\right|}$$
where $G_{d}$ and $P_{d}$ are the sets of all pixels within a d-pixel distance from the ground truth and prediction mask. d equals 2% of an instance diagonal (15-pixel distance on average).

To quantitatively assess the accuracy of amodal detection, we measured the (IoU) value between the amodal prediction of occluded mask $P_{a}$ against the ground truth (GT) occluded grape mask $G_{a}$, named amodal IoU.(13)$$\mathrm{amodal}\;\mathrm{IoU}\;(G_{a},\;P_{a})=\frac{\left|G_{a}\cap P_{a}\right|}{\left|G_{a}\cup P_{a}\right|}$$

The GT occluded grape masks were obtained by annotations for manual de-occlusion images (Figure 7). A higher amodal IoU value closer to 1 indicates more accurate detection that is aligned with the ground truth.

To facilitate the training of the grape amodal detection model, a dataset comprising 2427 complete grape instances was established, which were then loaded into the dataloader and generated synthesis images (320 × 320 pixel) before being fed into the network in a batch size of 16. The U-Net serves as the feature extractor for the grape amodal detection model. The networks were not initialized with any pretrained weights, and the weights of the networks were optimized using an Adam optimizer. The learning rate, momentum, and weight decay are set to 0.0001, 0.9, and 0.0005, respectively. The grape amodal detection model was trained for 15,000 iterations.

## 4. Experiment Results

### 4.1. Impact of Segmentation Performance on Amodal Detection

We compared the grape instance segmentation model with the existing mainstream instance segmentation methods. We compared the model with the two-stage segmentation methods Mask R-CNN [24], BMask R-CNN [22], Pointrend [19], and Transfiner [21] and the single-stage methods SOLOv2 [28] and YOLACT++ [29]. In addition, we also report the amodal detection performance based on results of these segmentation methods on a grape amodal detection dataset.

As shown in Table 2, we conducted a comparative analysis of our method on the grape instance segmentation dataset. Our algorithm achieved the best performance in terms of segmentation average precision (AP). Notably, on the boundary segmentation evaluation metric AP^B^, our method demonstrated a 3.9% performance improvement compared to the second-best method, Transfiner. Although our method achieved only a modest 0.8% improvement in AP compared to the second-best segmentation method, SOLOV2, it demonstrated a significant 9.12% enhancement in the amodal detection performance metric (amodal IoU). This indicates that our proposed instance segmentation method offers considerable advantages in grape amodal detection tasks.

To provide a more intuitive comparison, we visualized the segmentation and amodal detection results of all the comparison methods, as shown in Figure 8. From the visualization results, it can be seen that the baseline segmentation method, Mask-RCNN, exhibits significant misalignment in the segmentation boundaries of the overlapping grape regions, which leads to its corresponding non-pattern detection performance (only 0.5072 and 0.7313) being significantly lower than the other segmentation methods. Compared to other methods, our segmentation method achieves better boundary alignment results in the overlapping grape regions. Compared to the Pointrend segmentation method, under similar amodal IoU values before amodal prediction (0.4657, 0.4032 vs. 0.4625, 0.3934), our method achieves a significantly higher amodal IoU value (0.8762, 0.8421 vs. 0.8031, 0.8009) after amodal prediction.

Although SOLOV2 achieved the second-highest AP score (0.890 in Table 2) in the segmentation challenge, its performance in amodal prediction was relatively mediocre (0.7019). This is primarily due to its inability to produce well-aligned boundaries when segmenting overlapping regions between occluded and occluding instances. In comparison, BMask R-CNN achieves higher amodal detection performance (0.8442 vs. 0.6851 in Figure 8) despite a slightly lower segmentation amodal IoU (0.3914 vs. 0.3971 in Figure 8). This improvement can be attributed to its more tightly aligned segmentation boundaries. In contrast, our segmentation method provides even more precise boundary segmentation, especially in overlapping regions, where the boundaries between occluded and occluding instances are better aligned. This significantly enhances the amodal detection performance of our method.

### 4.2. Ablation Study

To gain a deeper understanding of how the incoherent decoder and pixel affinity learning module guide grape instance segmentation, we performed a related ablation experiment, as shown in Table 3. We made the following observations:Compared to using the boundary decoder to guide the instance segmentation, embedding our incoherent region decoder delivers comparable performance in terms of the instance segmentation metric AP under the same decoder output size conditions. However, the incoherent decoder with its decoder output set to 28 provides 2.6% improvements in boundary segmentation metrics AP^B^.Whether using the boundary decoder or the incoherent region decoder, incorporating the pixel affinity learning module can significantly improve the segmentation method performance in terms of both instance segmentation metrics and boundary evaluation metrics.Whether using the boundary decoder or the incoherent region decoder, adopting a larger decoder output size (56) can achieve better instance segmentation performance compared to when the decoder output was set to 28.

To provide a more intuitive comparison, we visualized the masks, incoherent regions, or boundary predictions produced by different segmentation methods, as shown in Figure 9. Compared to the BMask-RCNN method integrated with the pixel affinity learning module, it becomes evident that the absence of this module limits the ability to consistently translate boundary predictions into the corresponding mask prediction results (blue line), resulting in misalignment between mask predictions and boundary predictions. Furthermore, the boundary decoder often produces discrete predictions in regions where instances overlap. By replacing it with our incoherent decoder, more accurate incoherent region predictions in overlapping areas (green line) can be observed, exhibiting higher consistency with the mask prediction results.

The overlapping cover strategy is proposed to synthesize occluded grapes, ensuring the realism of the synthetic instances and thereby enhancing amodal detection performance. To validate the impact of our strategy, we conducted comparative experiments. The quantitative outcomes are detailed in Table 4. Moreover, the comparative visualization results are presented in Figure 10. We made the following observations:Whether based on annotations or instance segmentation results, amodal detection performance shows a significant improvement in amodal IoU metrics (0.6231 vs. 0.8991, 0.5740 vs. 0.7931).Compared with the random cover strategy [1,15,16], adopting the proposed overlapping cover strategy exhibits superior amodal detection performance in terms of annotations (0.8991 vs. 0.7933) and instance segmentation results (0.7931 vs. 0.6838).There is a notable amodal detection performance gap between the annotation-based and segmentation-based predictions (10.60% by the overlapping strategy, 10.95% by the random cover strategy) that can be witnessed. Compared to the segmentation results, the annotation instance pair shows better alignment of overlapping boundaries, further emphasizing the importance of high-quality segmentation for subsequent amodal detection.The visualization results shown in Figure 10 indicate that, compared to the random cover strategy, the amodal detection model adopting the overlapping cover strategy is better at detecting the potential contours of occluded grapes.

When using cross-prediction to distinguish occluder and occluded instances, the decision strategy used in (1) achieves higher accuracy on the condition that the $\Delta_{M}$ value for occluder instances is lower. Therefore, we introduced a reverse synthesis factor γ into our synthesis strategy. As illustrated in Figure 6a, reverse synthesis inverts the roles: both $I_{Occ}^{s}$ and $I_{\mathrm{Intact}}^{s}$ are treated as occluder instances, while $I_{\mathrm{Occder}}^{s}$ is designated as the occluded instance. This approach incorporates a certain proportion of equivalent mappings into the learning process $f(I_{Occ}^{s},I_{\mathrm{Occder}}^{s})\Rightarrow I_{\mathrm{Intact}}^{s}$, effectively regularizing amodal detection model and mitigating its tendency to over-predict occluder instances. Additionally, during the instance synthesis process, as outlined in Algorithm 1, we also incorporated a size-matching mechanism for occluded and occlude instances, along with an occlusion rate detection mechanism, to minimize distortion in the synthesized images.

To quantitatively analyze the impact of the above factors on amodal detection performance, we conducted the relevant ablation experiments. The results, reported in Table 5, reveal the following insights:The quantitative results demonstrate that excluding reverse synthesis factor has no influence on the prediction of occluded instances (0.7931 vs. 0.7958). However, it has a significant negative impact on the prediction of occluded instances (0.9469 vs. 0.8271), leading to higher $\Delta_{M}$ values.The quantitative results report a negative effect of amodal detection performance of occluded instances in the absence of size and occlusion rate mechanisms.

Figure 11 illustrates the distorted synthesis images resulting from the absence of size matching and occlusion detection mechanisms. When there is a significant size discrepancy between the occluder and occluded instances, the occluder instance is fully incorporated into the synthesized result of the occluded instance, which clearly deviates from realistic grape occlusion scenarios. Additionally, a smaller occlusion rate β during the synthesis process will lead to a synthesis image without occlusion, emphasizing the importance of the filtering mechanism. Moreover, Figure 11 also presents the effect of the reverse synthesis strategy in mitigating the over-prediction of occluder instances by amodal detection model.

Some previous amodal detection research [1,32] reports that generative adversarial networks (GANs) outperform generative networks concerning amodal detection performance. To expand upon our proposed synthesis method, we trained a GAN and conducted a quantitative comparison. The results, showcased in Table 6, indicate that our model achieved strong amodal detection performance. This emphasizes the effectiveness of the generative network in effectively addressing the grape amodal detection task.

## 5. Discussion

### 5.1. Summary of the Experiment Results

The proposed instance segmentation method achieved the best performance on the grape instance segmentation dataset, significantly surpassing other methods in boundary segmentation evaluation metrics. Furthermore, based on the segmentation results, it also delivered optimal performance in grape amodal detection (Table 2).The visualization results show that the proposed instance segmentation method more effectively distinguishes overlapping regions of grape instances, which is critical for grape amodal detection (Figure 8).As presented in Table 4, the comparison results demonstrate that the proposed overlapping cover strategy outperforms the random cover strategy, achieving superior performance in grape amodal detection.

### 5.2. Benefits of Grape Amodal Detection

In image-based yield estimation for grapevines, metrics such as grape cluster counts [9], berry counts [1], and identified grape areas [5,6] serve as crucial predictors. However, occlusion factors can cause the berry count and identified areas to be significantly underestimated compared to the actual values. Our proposed amodal detection method offers a more intuitive approach to addressing occlusion, as opposed to regression correction methods. It is worth noting that our method and regression correction are not mutually exclusive; instead, their combination produces more accurate results when observations exhibit smaller deviations from the ground truth.

### 5.3. Why Does the Overlapping Cover Strategy Work Well?

In the self-supervised training of the amodal detection model, the alignment between synthetic domain and real domain plays a crucial role in determining amodal detection performance. The random occlusion strategy for synthesizing occluded grapes often generates noticeably distorted synthetic cases (Figure 2b). To address this, we propose an overlapping cover strategy that ensures the synthetic occluded grapes closely resemble real-world scenarios. Additionally, a size matching and occlusion rate detection mechanism is embedded within this strategy to maximize the realism of the synthetic instances while minimizing distortion.

### 5.4. Why Does the Segmentation Method Need to Focus on Overlapping Areas? Why Does Our Instance Segmentation Method Work Well?

Both the visualization (Figure 8) and quantitative comparison results (Table 2) indicate that even minor segmentation differences in overlapping instance regions can lead to significant performance disparities in subsequent grape amodal detection.

In the proposed segmentation algorithm, we drew inspiration from BMask R-CNN [22] by introducing an additional detector to supervise the error-prone areas during the segmentation process. Unlike methods that focus on instance boundaries, our approach targets the regions lost during the size transformation of instances, enabling it to adapt more flexibly to varying instance sizes (Figure 4). Leveraging the profound correlation between instance masks and incoherent region masks, we introduced a pixel affinity learning module positioned between these branches to facilitate their interconnected learning. Visual representations from ablation comparative experiments (Figure 9) demonstrated the module’s effectiveness in enhancing the collaboration between instance mask and incoherent region predictions.

### 5.5. Dilemmas and Limitations of Our De-Occlusion Method

Although the proposed grape amodal method demonstrates partial prediction capability in grape occlusion, the disparity between annotation-based and mask-based estimations underscores that the primary impediment to effective de-occlusion prediction lies in achieving more precise and accurate instance boundary segmentation results. Yet, even the SOTA instance segmentation method struggles to accurately delineate boundaries in complex areas like grape overlapping regions, characterized by strong structural ambiguity. Additionally, the improvement in segmentation performance always comes at the cost of inference speed (Table 2). Although grape amodal detection places more emphasis on offline processing to achieve higher accuracy and is less sensitive to inference speed, developing lightweight segmentation algorithms for easier deployment remains one of the key areas for future work.

Regarding amodal prediction, although annotation-based amodal detection achieved an amodal IoU value of 0.8991, the excessive smoothing of detection region boundaries starkly distinguishes them from reality (Figure 10). Real-world scenarios often involve instances where one grape is obscured by multiple grapes. However, synthesizing grape instances with multiple occluders poses challenges within the synthesis algorithm, limiting the amodal for these occluded grapes.

## 6. Conclusions

To achieve amodal detection for grapes, we propose a grape instance segmentation model and a grape amodal detection model. Specifically, to reduce the impact of segmentation gaps on subsequent amodal detections, our grape instance segmentation model integrates an additional incoherent region decoder to focus on error-prone areas caused by size transformations during training, particularly in regions of overlapping instances. Additionally, a pixel affinity learning module is introduced to harmonize the predictions of the mask decoder with incoherent region prediction. For amodal detection, a novel overlapping cover strategy is introduced to generate training data that closely mirror real-world grape occlusion scenarios, significantly improving the amodal detection performance. The experimental results on the grape amodal detection dataset demonstrate that the proposed segmentation method outperforms comparison methods in amodal detection performance, achieving an amodal IoU of 0.7931. Additionally, compared to the random cover strategy, the proposed overlapping cover strategy achieves superior amodal detection performance. The advantages brought by our method can be summarized as follows:Our grape instance segmentation model delivers more refined grape instance segmentation results, particularly in overlapping regions.Fine-grained segmentation in overlapping regions contributes to improved performance in subsequent amodal detection.The proposed overlapping cover strategy ensures distortion-free synthetic data, thereby reducing the disparity between the synthetic domain and the real-world occlusion domain.

## Figures and Tables

**Figure 1 sensors-25-01546-f001:**
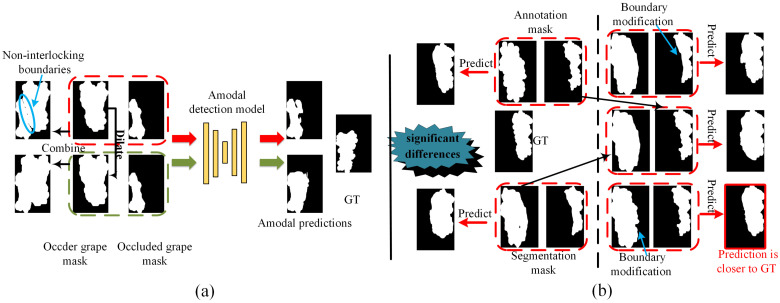
The impact of instance segmentation performance on amodal prediction. (**a**) The segmentation gap between occluder and occluded instances significantly affects the performance of amodal prediction. The term “dilate” refers to the dilation operation in image processing, which expands the boundaries of occluder masks. (**b**) Comparison of amodal detection performance across different input instance pairs. “Boundary modification” refers to using the overlapping boundaries of the occluded or occluder instance to adjust the boundaries of another instance.

**Figure 2 sensors-25-01546-f002:**
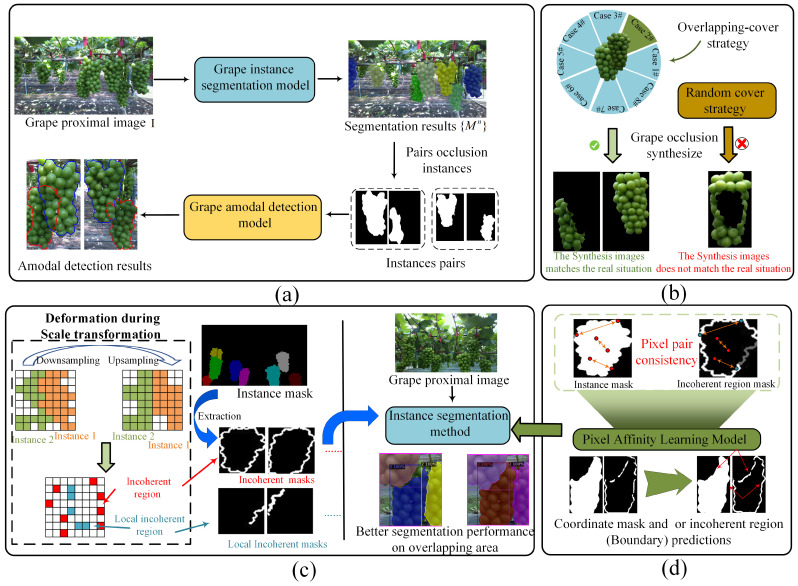
An overview of our work. (**a**) The process of grape amodal detection. (**b**) A novel overlapping cover strategy in our grape amodal detection model suitable for grape occlusion to replace the random cover strategy. (**c**) Utilizing incoherent regions to improve grape instance segmentation performance in overlapping areas. (**d**) Utilizing pixel affinity learning model to coordinate segmentation and incoherent predictions. Since the incoherent region mask is a subset of the instance mask, there exists a correlation between the pixels at the same positions in both masks.

**Figure 3 sensors-25-01546-f003:**
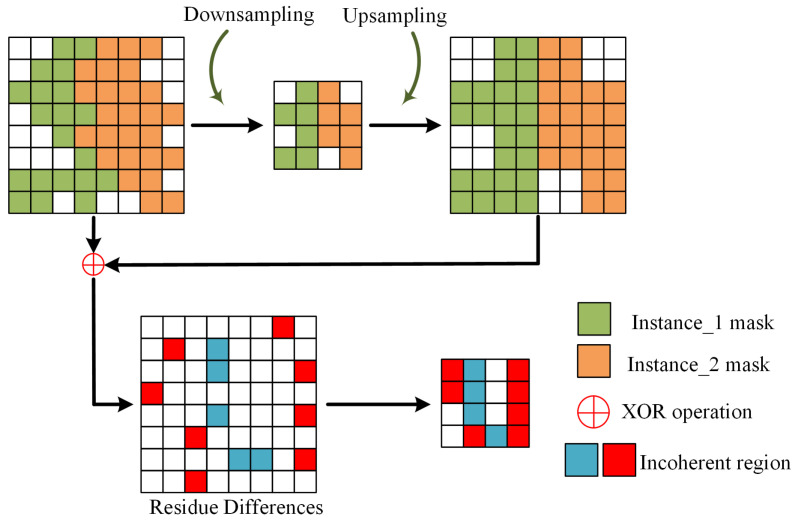
Incoherent regions distributed along instance boundaries and within overlapping areas.

**Figure 4 sensors-25-01546-f004:**
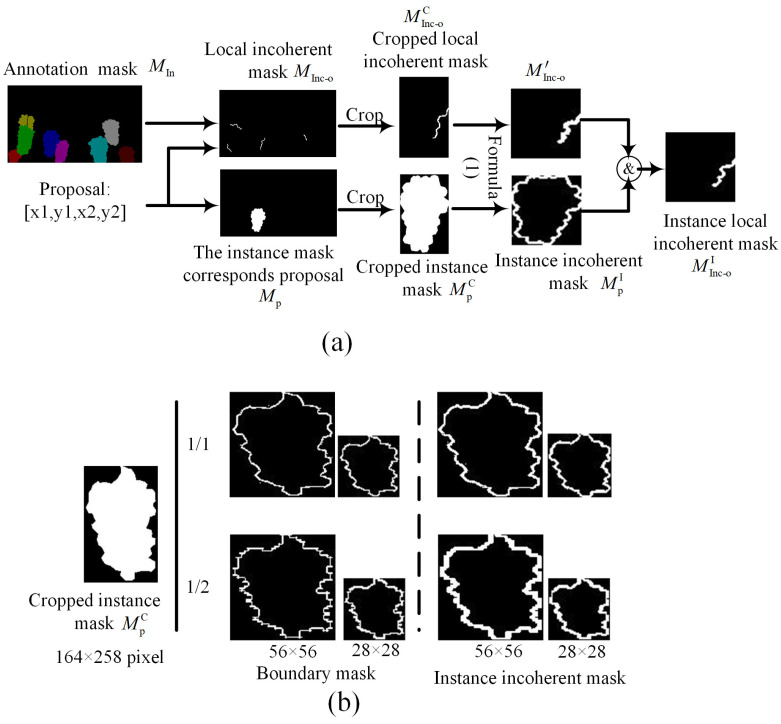
(**a**) The extraction process of instance incoherent and local incoherent region masks according to a proposal. (**b**) The impact of different output sizes on boundary masks and instance incoherent region masks. The boundary masks are derived by applying the Laplacian operator to the instance masks. The “1/2” indicates the boundary and incoherent masks extracted when the instance masks are resized to half of their original size.

**Figure 5 sensors-25-01546-f005:**
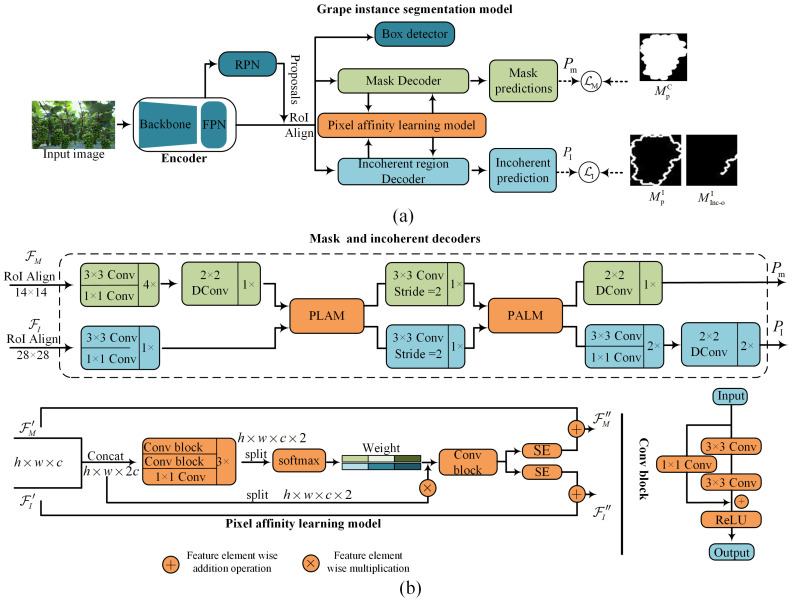
The grape instance segmentation model. (**a**) An overview of the grape instance segmentation model. (**b**) The detailed structure of the mask decoder and incoherent region decoder. The notation “3 × 3 Conv” refers to a convolution operation using a 3 × 3 kernel with a stride of 1. “Stride = 2” indicates that the convolution operation uses a stride of 2. “Dconv” represents a deconvolution operation, which is used for upsampling the image. “n×” signifies the number of times the current module is repeated. “Concat” means concatenating two features along the channel dimension, while “split” refers to splitting the features into two equal parts along the channel dimension. “Softmax” is an activation function applied to the input. “SE” denotes the squeeze-and-excitation channel attention module.

**Figure 6 sensors-25-01546-f006:**
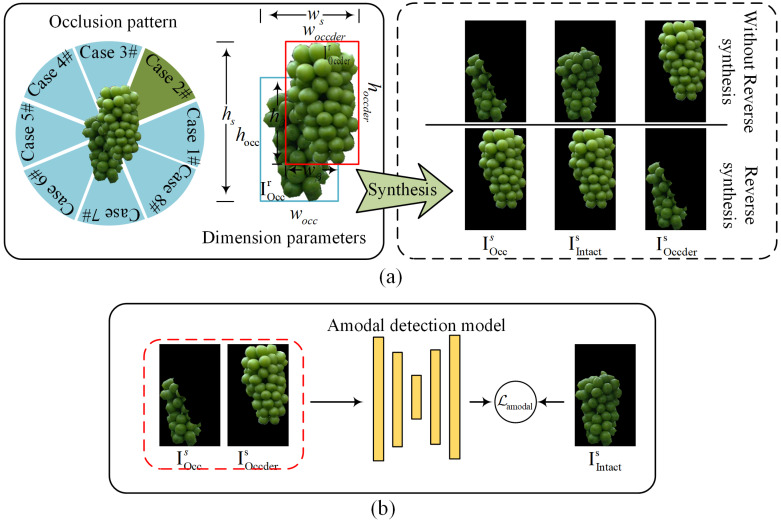
(**a**) The definition and synthesis output of the overlapping cover strategy. (**b**) The training of the grape amodal detection model. The ‘dial’ represents the direction angles of the 8 occlusion modes. Specifically, it represents how the occluder instance covers the occluded instance from 8 different directions in the synthesis method. Taking Case #2 as an example, the occluder instance covers the occluded instance within the angular range of [22.5°, 67.5°].

**Figure 7 sensors-25-01546-f007:**
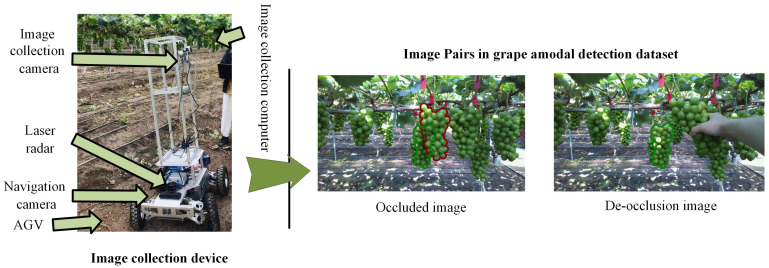
Image collection device and image pairs in the grape amodal detection dataset. An image pair includes an occluded image and the corresponding de-occluded image.

**Figure 8 sensors-25-01546-f008:**
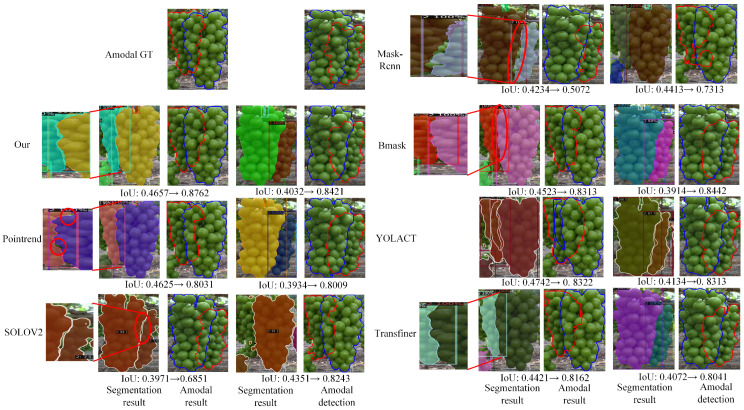
A visualization of the segmentation results and the corresponding amodal detection results. The A and B in the expression ‘IoU: A→B’ represent the amodal IoU values between the occluded instance and the GT of the occluded instance before and after amodal detection. In the segmentation visualization, YOLOACT did not use the NMS method. We selected the segmentation result with the highest IoU value between the predicted and ground truth amodal masks for amodal prediction.

**Figure 9 sensors-25-01546-f009:**
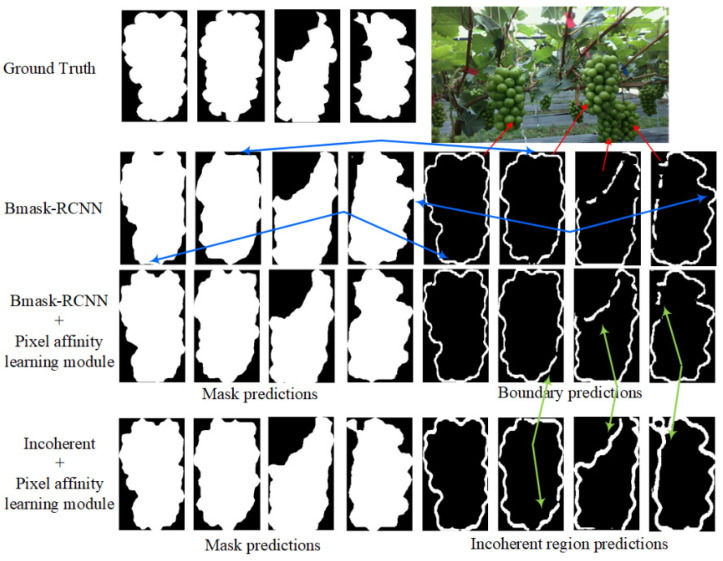
Visualization of mask, boundary, and incoherent region predictions. The blue arrows indicate the misaligned regions between the mask prediction results and the boundary prediction results in Bmask-RCNN.

**Figure 10 sensors-25-01546-f010:**
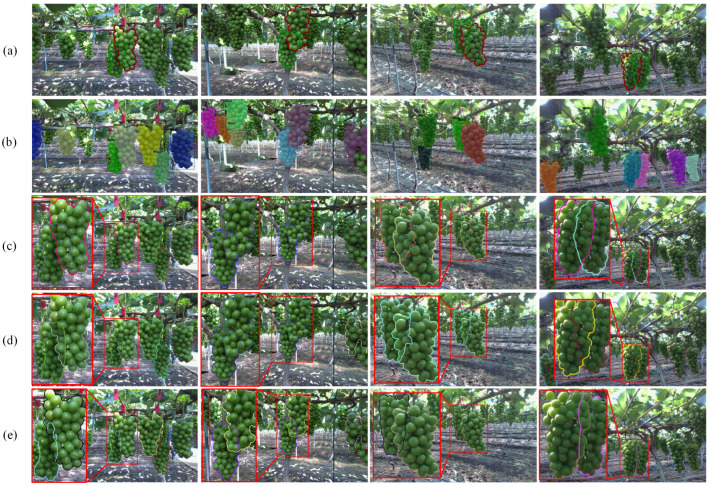
A visualization comparison of grape amodal detection on the grape amodal detection dataset. (**a**) Amodal annotations according to the image pairs, (**b**) segmentation results, (**c**) amodal detection predictions on the segmentation results by the grape amodal detection model, (**d**) amodal detection predictions on annotations by the grape amodal detection model, (**e**) amodal detection predictions on segmentation results by the random cover strategy.

**Figure 11 sensors-25-01546-f011:**
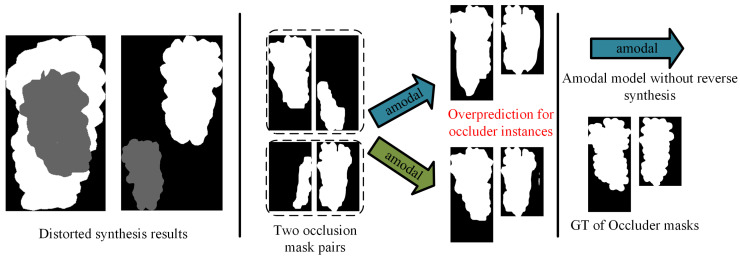
The distorted synthesis result occurs when there is no size matching and occlusion detection mechanism, and the over-prediction of the occluder instance happens in the absence of the reverse synthesis strategy.

**Table 1 sensors-25-01546-t001:** Detailed information of datasets.

Dataset	Dates	Training Set		Validation Set		Test Set	
Grape instance segmentation		Images	Instances	Images	Instances	Images	Instances
	23 June 2022	1236	7183	406	2355	406	2419
	3 June 2023	---	---	---	---	97	679
	Total	1236	7183	406	2355	503	3098
Grape amodal detection	**Dates**	**Image Pairs**					
	23 June 2022	34					
	3 June 2023	136					
	Total	170					

**Table 2 sensors-25-01546-t002:** Performance comparison of different instance segmentation methods on grape instance segmentation dataset and grape amodal detection dataset.

Methods	AP	AP^Box^	AP^B^	FPS	Amodal IoU
YOLACT++	0.873	0.806	0.464	**52.1**	0.7334
SOLOv2	0.890	——	0.502	21.9	0.7019
Mask R-CNN	0.823	0.843	0.435	42.5	0.6211
BMask R-CNN	0.863	0.817	0.466	37.5	0.7445
Pointrend	0.889	0.836	0.506	30.4	0.7276
Transfiner	0.752	0.857	0.267	3.22	—
Transfiner ^2×^	0.877	0.861	0.521	3.22	—
Transfiner *	0.882	0.859	0.521	3.22	0.7233
Our	0.898	0.859	0.560	21.2	0.7931

‘^2×^’ indicates training with 2× iterations; ‘*’ indicates use of the model weight-trained by our method instead of the ImageNet weight to initialize the backbone and RPN of the transfiner model. FPS means frames per second, a metric used to evaluate the processing speed.

**Table 3 sensors-25-01546-t003:** Performance ablation of grape instance segmentation methods.

Index	Detector	PLAM	Output Size	AP	AP^Box^	AP^B^	FPS
1	×	×	----	0.823	0.843	0.435	42.5
2	Boundary	×	28	0.863	0.817	0.466	37.5
3	Boundary	×	56	0.887	0.851	0.534	16.0
4	Boundary	√	28	0.889	0.841	0.527	27.4
5	Boundary	√	56	0.889	0.843	0.537	19.5
6	Incoherent	×	28	0.865	0.822	0.492	37.5
7	Incoherent	√	28	0.889	0.846	0.549	27.4
8	Incoherent	√	56	0.898	0.859	0.560	21.2

√, × Indicates whether to adopt or not to adopt this module.

**Table 4 sensors-25-01546-t004:** Quantitative results of grape amodal detection.

	Synthesis Strategy	Amodal IoU
On annotations	---	0.6231
	Overlapping cover	0.8991
	Random cover	0.7933
On segmentations	---	0.5740
	Overlapping cover	0.7931
	Random cover	0.6838

**Table 5 sensors-25-01546-t005:** Quantitative results of size mechanism and reverse synthesis.

Rate and Size	Reverse Synthesis	Occluded Amodal IoU	Occluder IoU
√	√	0.7931	0.9469
×	√	0.7649	0.9413
√	×	0.7958	0.8271

√, × Indicates whether to adopt or not to adopt this method.

**Table 6 sensors-25-01546-t006:** Quantitative results of different amodal detection models.

		Occluded Amodal IoU	Occlude IoU
Ours	On annotation	0.8991	0.9763
GANs	On annotation	0.8478	0.9319
Ours	On segmentation	0.7931	0.9469
GANs	On segmentation	0.7752	0.9357

## Data Availability

The related data are available at https://doi.org/10.5281/zenodo.14630616, accessed on 2 July 2023.

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
