# Peer review of "Incoherent Region-Aware Occlusion Instance Synthesis for Grape Amodal Detection"

_sensors, 2025, doi:10.3390/s25051546_

Round 1
Reviewer 1 Report
Comments and Suggestions for Authors
The manuscript presents a method for detecting two grape clusters that partially overlap in a photograph, separating them, and reconstructing the missing parts.
Overlapping of grape bunches on top of each other in photos of real vineyards is a problem.
This phenomenon is very common and makes it impossible to accurately predict the amount of future harvest. In addition, I believe that in robotic grape harvesting this phenomenon will affect the positioning accuracy of the cutting tools.
From this we can conclude that the research of the authors of the manuscript has a high practical significance.
However, I have questions for the authors of the manuscript.
Please explain the meaning of the dial in Figure 6.
Please write in detail what you understand by IOU. Explain how you got 4 significant digits after the decimal point.
Please explain the data in Tables 2 and 3 in more detail. The header presents a number of indicators that need further explanation.
You write that Figure 8 clearly demonstrates the results. Unfortunately, I do not think so. Please describe it in more detail.
What do the blue arrows in figure 9 mean?
The inscriptions in most of the figures are very small.
Unable to open link to github.com
Reviewer 2 Report
Comments and Suggestions for Authors
We recognize the value of an incoherent region-aware occlusion instance synthesis for grape amodal detection.
Observations:
1. Key contributions of the authors include the following:
a. Grape Instance Segmentation Model: error-prone, especially in areas with overlapping instances.
b. Amodal Detection method: Notable IoU enhancement (0.7931).
c. Self-supervised Learning approach: Utilizing synthetic occlusion to generate training data for the modal detection method.
2. The methodologies and results in Section 4 were effectively presented with informative visuals, both experiment and real time implementation.
Suggestions and Queries
1. The authors' contributions are efficiently outlined in lines 97-110 and 111-122, to avoid redundant information.
2. This approach did not employ explicit NMS to eliminate duplicates.
3. Depth-cue details were not included in this study for any reason.
4. Although the pixel affinity learning module enhanced segmentation, it did not address the computational cost and power consumption.
